# *PagbHLH35* Enhances Salt Tolerance through Improving ROS Scavenging in Transgenic Poplar

**DOI:** 10.3390/plants13131835

**Published:** 2024-07-03

**Authors:** Shuang Wang, Liben Dong, Wenjing Yao, Tingbo Jiang

**Affiliations:** 1State Key Laboratory of Tree Genetics and Breeding, Northeast Forestry University, Harbin 150040, China; wangsdemon@163.com (S.W.); hljs_dlb@126.com (L.D.); yaowenjing@njfu.edu.cn (W.Y.); 2Co-Innovation Center for Sustainable Forestry in Southern China, Nanjing Forestry University, 159 Longpan Road, Nanjing 210037, China; 3Harbin Research Institute of Forestry Machinery, State Administration of Forestry and Grassland, Harbin 150086, China; 4Research Centre of Cold Temperate Forestry, Chinese Academy of Forestry, Harbin 150086, China

**Keywords:** *Populus alba* × *P. glandullosa*, bHLH transcription factor, salt stress, ROS scavenging, transgenic poplar

## Abstract

The bHLH transcription factor family plays crucial roles in plant growth and development and their responses to adversity. In this study, a highly salt-induced bHLH gene, *PagbHLH35* (Potri.018G141600), was identified from *Populus alba × P. glandullosa* (84K poplar). PagbHLH35 contains a highly conserved bHLH domain within the region of 52–114 amino acids. A subcellular localization result confirmed its nuclear localization. A yeast two-hybrid assay indicated PagbHLH35 lacks transcriptional activation activity, while a yeast one-hybrid assay indicated it could specifically bind to G-box and E-box elements. The expression of *PagbHLH35* reached its peak at 12 h and 36 h time points under salt stress in the leaves and roots, respectively. A total of three positive transgenic poplar lines overexpressing *PagbHLH35* were generated via *Agrobacterium*-mediated leaf disk transformation. Under NaCl stress, the transgenic poplars exhibited significantly enhanced morphological and physiological advantages such as higher POD activity, SOD activity, chlorophyll content, and proline content, and lower dehydration rate, MDA content and hydrogen peroxide (H_2_O_2_) content, compared to wild-type (WT) plants. In addition, histological staining showed that there was lower ROS accumulation in the transgenic poplars under salt stress. Moreover, the relative expression levels of several antioxidant genes in the transgenic poplars were significantly higher than those in the WT. All the results indicate that *PagbHLH35* can improve salt tolerance by enhancing ROS scavenging in transgenic poplars.

## 1. Introduction

Plants frequently face various environmental stressors including drought, salinity, extreme temperature, and heavy metal pollution in natural condition [1,2]. Among these, salinization particularly affects agricultural and forestry production as well as ecosystems. It is projected that around 1 billion hectares of soil are conditioned using salinization worldwide, with the area of saline-alkali land in China almost reaching 100 million hectares [3,4]. Cultivating salt-tolerant plant varieties is of significant importance for the effective utilization of saline-alkali land resources and ecological environmental protection.

To evolve in the face of these environmental stresses, a variety of sophisticated biological adaptations have emerged in plants at the levels of form, function, and genetics [5,6,7,8]. In particular, transcription factors (TFs) play crucial roles in response to environmental stresses through transcriptional regulation in plants [9]. Widely distributed across plants, animals, and fungi, the bHLH TF family stands as a major class of transcription factors within the plant domain. An expanding array of research has demonstrated the key roles of bHLH TFs in growth, metabolic processes, and the capacity to withstand negative environmental impacts [10]. In Arabidopsis, three bHLH TFs including *AtbHLH92*, *AtbHLH17* (*AtAIB*), and *AtbHLH122* regulate plant responses using the ABA-regulated mechanism to counteract the effects of drought, salinity, osmotic stress, oxidation, and cold stress [11,12,13]. *AtbHLH92* can affect salt stress adaptation through the regulation of stress-responsive gene expression and *AtbHLH122* functions as an enhancer of drought tolerance, salt tolerance, and osmotic signaling [11,14]. The overexpression of grapevine *VvbHLH1* significantly enhances salt and drought tolerance in transgenic Arabidopsis [15]. *CmbHLH1* boosts iron uptake in Chrysanthemum by enhancing the transcription of genes responsive to iron ions [16]. *OsbHLH148* enhances plant drought resistance by interacting with OsJAZ proteins involved in jasmonic acid signaling in rice [17]. *OrbHLH2*, when excessively expressed in Arabidopsis, leads to an increase in the transcription of stress-indicative genes like *DREB1A/CBF3*, *RD29A*, *COR15A*, and *KIN1*, which enhances the salt and osmotic resistance of the genetically modified plants [18].

There is a total of 202 bHLH family members in *Populus trichocarpa*. Although bHLH TF members have been confirmed within the model botanical species including Arabidopsis and rice, it is less known in the functions and modes of action of the bHLH family member in poplar. The bHLH TF family was explored in depth in a previous study to further verify our previous findings [10]. In the study, we identified that *PagbHLH35* in 84K poplar showed a significantly high expression after salt stress using RT-qPCR, which reached its peak at 12 h and 36 h time points in the leaves and roots, respectively. Experiencing salt stress, the engineered poplar lines demonstrated enhanced morphological and physiological properties relative to the WT. There was lower ROS accumulation detected in the transgenic poplars under salt stress. In addition, the relative expression levels of several antioxidant genes in the transgenic poplars were significantly higher than those in the WT. Moreover, *PagbHLH35* was identified to specifically bind to G-box and E-box. All the results indicated *PagbHLH35* may improve salt tolerance by regulating antioxidant genes through specifically binding to G-box and/or E-box, which provides a reference for the further understanding of plant adaptation to adverse conditions and the molecular cultivation of plants with high salinity tolerance.

## 2. Materials and Methods

### 2.1. Plant Materials and Methods

We collected small branches from *Populus alba × Populus glandulosa* plants grown in the greenhouse, maintaining relative humidity between 60 and 70%, alternating 16 h of light with 8 h of darkness, and maintaining a mean temperature of 25 °C. These branches were placed in the same beaker and cultured in water until new shoots and roots sprouted. Then, we selected 24 plants with similar growth conditions, and randomly divided them into two groups, with three biological replicates per group. Subsequently, one group was subjected to 150 mM NaCl for 0 h, 12 h, 24 h, and 36 h, and another group was placed in water as control. Samples of roots and leaves from each group were swiftly frozen with liquid nitrogen and preserved at −80 °C in a freezer for subsequent RT-qPCR analysis.

The leaves from the tissue culture seedlings were placed on differentiation medium for 30 days, then the fresh shoots were transferred to rooting medium (Woody Plant Medium) for 30 days. Genetic transformation was carried out on one-month-old histocultured seedlings.

### 2.2. Sequence Analysis of PagbHLH35

Protein sequences of PagbHLH35 and homologous genes from different species were obtained from NCBI (https://blast.ncbi.nlm.nih.gov/Blast.cgi, accessed on 2 March 2022). The alignment of numerous sequences was achieved with ClustalW. The phylogenetic tree illustrating the kinship of homologous proteins was generated using MEGA11’s neighbor-joining method.

### 2.3. Subcellular Localization of PagbHLH35

The PagbHLH35 ORF, stripped of a stop codon, was inserted into the pBI121-GFP vector, with the CaMV35S promoter as its regulatory element (Figure 1A). The recombinant plasmid was then integrated into *Agrobacterium* GV3101. *Agrobacterium* solutions containing 35S-GFP and 35S-PagbHLH35-GFP were, respectively, infiltrated into tobacco leaves, which were then incubated in darkness for 24 h. Laser scanning confocal microscopy was employed to view fluorescence signals (LSM 800, Carl Zeiss, Jena, Germany).

### 2.4. Yeast Hybrid Assays

We validated the transcriptional activity of PagbHLH35 protein through yeast two-hybrid (Y2H) experiments [19]. The CDS of PagbHLH35 was incorporated into the pGBKT7 vector to construct the fusion vector pGBKT7-PagbHLH35. The fusion, negative control pGBKT7, and positive control pGBKT753/pGADT7-T were transformed into Y2H yeast strains. Positive yeast colonies were grown on medium without tryptophan, and β-galactosidase activity was assayed on medium without tryptophan and histidine, with X-α-Gal.

The yeast one-hybrid method was applied to verify whether PagbHLH35 can uniquely interact with G-box and E-box elements [20,21]. The G-box and E-box elements, each repeated three times, were integrated into the pAbAi vector to create the respective bait reporter constructs. The coding sequence (CDS) of the *PagbHLH35* gene was then fused into the pGADT7 vector, yielding the pGADT7-PagbHLH35 fusion construct. These constructs were co-transformed into Y1H yeast, and positive yeast colonies were identified on SD/-Leu medium and further confirmed on SD/-Leu/AbA medium with serial dilutions at 10×, 100×, and 1000×. 

### 2.5. The Acquisition and Identification of Transgenic Poplar Lines

We employed homologous recombination to construct a plant expressing vector of pBI121-PagbHLH35. The engineered vector was subsequently integrated into *Agrobacterium* GV3101. Excised leaves from one-month-old poplar tissue culture seedlings were immersed in the bacterial suspension for a duration of 10 min. Subsequently, these leaves were placed on a selective medium supplemented with 50 mg/L kanamycin and 200 mg/L cefotaxime, where they remained until shoots displaying resistance developed. These resistant shoots were then transferred to MS medium with the same antibiotic concentrations to cultivate resistant seedlings. Molecular identification was performed through extracting DNA from resistant seedling leaves using specific primers (Appendix A).

### 2.6. Histological Staining and Physiological Measurements

Three T_3_ transgenic poplar lines at one-month-old were treated with 150 mM NaCl. Post a four-hour treatment phase, the leaves were procured for staining with nitroblue tetrazolium (NBT) and 3,3′-diaminobenzidine (DAB) staining, followed by destaining with absolute ethanol. After treatment for seven days, the leaves were collected for physiological measurements with three biological replicates per treatment. Physiological indicators including malondialdehyde (MDA; A003-3-1), chlorophyll (A147-1-1), peroxidase (POD; A084-3-1), superoxide dismutase (SOD; A001-1-2), and hydrogen peroxide (A007-1-1) were quantified utilizing assay purchased from Jiangsu Jiancheng Bioengineering Institute, Nanjing, China.

### 2.7. The Expression Analysis of PagbHLH35 Gene and Antioxidant Genes

The total RNA was obtained through the use of an RNA extraction kit (TAKARA, Dalian, China), and cDNA was obtained using a reverse transcription kit (TAKARA, Dalian, China). RT-qPCR was carried out using the TB Green^®^ Premix Ex Taq™ II kit (TAKARA, Dalian, China) through qTOWER³ real-time detection system. The procedures were followed as described in our previous study [22]. The actin gene was the baseline for gene expression analysis. The relative levels were determined using the 2^−∆∆Ct^ method. Actin gene was used as the reference gene. The relative expression levels of genes were calculated using the 2^−∆∆Ct^ method [22]. All the primers were listed in Appendix A.

### 2.8. Statistical Analysis

The experiments were carried out on their own at least three times. The error bars correspond to the standard deviation. Student’s *t*-test was used for comparison of differences. *p* < 0.05 indicates statistical significance, denoted by *.

## 3. Results

### 3.1. Expression Pattern of PagbHLH35 Gene

The RT-qPCR method was applied to discern the expression dynamics of *PagbHLH35* in the roots and leaves in response to salt stress (all primers used in this study were listed in Appendix A). As shown in Figure 1B, the relative expression level of *PagbHLH35* peaked at the 12 h time point in the leaves and then gradually decreased under salt stress, while it slowly increased in the roots, reaching its highest expression level at the 36 h time point. It is speculated that *PagbHLH35* is involved in the response to salt stress (Figure 1B).

### 3.2. Characterizationof PagbHLH35 Gene

The *PagbHLH35* gene was successfully cloned from 84K poplar with specific primer pairs (Appendix A), which encodes 244 amino acids. A sequence analysis using the ProtParam tool [23] showed that the molecular formula of this protein is C_1216_H_1955_N_339_O_393_S_11_, with a relative molecular mass of 27.9646 kDa and theoretical isoelectric point (PI) of 5.23. The instability index of this protein is 49.49, classifying it as an unstable protein. The average hydrophilicity index is −0.586, and the aliphatic index is 81.15, indicating that this protein is hydrophobic. PagbHLH35 protein possesses a highly conserved bHLH (Basic-helix-loop-helix) domain, located between 71 and 114 amino acids (Figure 2A). Through BLASTP in NCBI https://blast.ncbi.nlm.nih.gov/Blast.cgi, accessed on 15 March 2022), 19 sequences with a high homology of *PagbHLH35* were obtained from various plant species, whose multiple sequence alignment was conducted using BioEdit version 7.2. Among them, the amino acid sequence of PagbHLH35 showed similarities of 99.59%, 95.08%, 89.3%, 70.46%, 69.77%, 64.37%, 64.08%, and 63.52% with the proteins from *Populus alba*, *Populus trichocarpa*, *Populus euphratica*, *Salix koriyanagi*, *Populus tomentosa*, *Manihot esculenta*, *Salix purpurea*, *Herrania umbratica*, *Jatropha curcas*, respectively. Phylogenetic analysis revealed that the gene *PagbHLH35* shares the highest degree of sequence similarity with XP_034902167.1, a protein from the white poplar (*Populus alba*). This implies that, among the species or genes analyzed, PagbHLH35 and XP_034902167.1 are evolutionarily the most closely related, suggesting a common ancestry or a recent divergence event between the two (Figure 2B).

### 3.3. PagbHLH35 Protein Was Localized in the Nucleus

To ascertain the intracellular distribution of the PagbHLH35 protein, we injected the recombinant vectors carrying 35S:PagbHLH35-GFP into tobacco leaf cells using *Agrobacterium*-mediated transient transformation. The results showed that the fluorescence signal of 35S:PagbHLH35-GFP vectors appeared manifested in the nucleus exclusively. In contrast, the fluorescence signal of control vectors was observed in both the nucleus and cytoplasm. This indicates that PagbHLH35 was localized in the nucleus (Figure 3A).

### 3.4. Transcriptional Activity of PagbHLH35 Protein

The negative control (pGBKT7), positive control (pGBKT7-53/pGADT7T), and PagbHLH35-BD yeast strains grew normally on SD/-Trp medium. However, the PagbHLH35-BD strain did not grow on SD/-Trp/-His/X-α-Gal medium, and did not exhibit blue coloration. This suggests that the PagbHLH35 protein lacks transcriptional self-activation activity (Figure 3B).

### 3.5. PagbHLH35 Specifically Binds to G-Box and E-Box Elements

As demonstrated in Figure 4, the positive control, negative control, and the yeast strains co-transformed with pGADT7-PagbHLH35 and pAbAi-Ebox or pAbAi-Gbox showed normal development on the SD/-Ura/-Leu medium. However, the yeast strains co-transformed with PagbHLH35 and pAbAi-Ebox or pAbAi-Gbox, as well as the positive control, grew normally on SD/-Ura/-Leu/AbA medium, while the negative control did not grow. This demonstrates that PagbHLH35 can specially bind to both E-box and G-box elements.

### 3.6. Molecular Identification of Transgenic Poplar Plants

A collection of three poplar lines, engineered to overexpress *PagbHLH35*, was procured. The transgenic poplar shoots were placed into the rooting media containing 50 mg/L kanamycin for positive screening. And the poplar seedlings capable of normal rooting (Figure 5C) were selected for PCR identification with specific primers (Appendix A). The relative expression levels of *PagbHLH35* transcripts were quantified using RT-qPCR in each line (Figure 5A). A significant increased expression of *PagbHLH35* in the overexpressing lines (Figure 5B) confirmed the successful transformation of the gene.

### 3.7. Salt Tolerance of Transgenic Poplar Overexpressing PagbHLH35 Gene

To prove the salt tolerance of the poplar with the overexpression of *PagbHLH35*, the transgenic seedlings at one-month-old were irrigated with 150 mM NaCl for 7 days. As illustrated in Figure 6A there was no significant difference in morphological growth between transgenic poplars and WT plants under normal conditions. Affected by salt stress, the WT poplar exhibited significant growth inhibition, whose leaves wilted severely and curled. In contrast, the leaves of transgenic poplar lines remained erect, with partial salt spots on the leaf edges (Figure 6A). In addition, the dehydration rate of the leaves in the transgenic poplars was 7.5% lower than that of WT plants after salt treatment for 7 h. The morphology advantage indicated that *PagbHLH35* participates in the positive regulation of salt stress response in transgenic poplars.

With standard environmental factors, there was no considerable change in the level of chlorophyll. However, after salt stress, the chlorophyll content of three transgenic lines was 6.63%, 6.56%, and 5.97% higher than that of WT, respectively (Figure 6C). After salt stress, the proline content of three transgenic lines, respectively, increased by 28.38%, 70.63%, and 48.18%, compared to the WT plants (Figure 6D). Under normal conditions, the transgenic and WT lines demonstrated an equivalent MDA content with no pronounced differences. After salt stress, the MDA content of both the WT and transgenic lines increased to varying degrees. Notably, the MDA content of transgenic poplars was significantly lower than that of WT plants, with a reduction of 14.53%, 25.86%, and 18.47%, respectively. These results indicate that the degree of membrane lipid peroxidation in overexpressing lines was lower than that in WT plants after salt stress, resulting in less damage from reactive oxygen species (ROS), thereby improving salt and osmotic stress tolerance (Figure 6E).

To verify whether the difference in MDA content between transgenic poplars and WT plants under salt stress is caused using ROS, we further measured the hydrogen peroxide (H_2_O_2_) content in their leaves. Under normal conditions, the accumulation levels of H_2_O_2_ in transgenic lines and WT were similar. However, with the treatment of 150 mM NaCl, the H_2_O_2_ content in transgenic poplars was significantly decreased by 35.23%, 30.38%, and 32.28%, respectively, compared to WT (Figure 6F). These results indicate that the substantial divergence in MDA content between the transgenic lines and WT resulted from the different accumulation levels of hydrogen peroxide generated after salt stress.

Antioxidant enzymes can scavenge ROS, thereby alleviating ROS damage caused by salt stress in plants. POD and SOD are important antioxidant enzymes that play a crucial role in protecting cells from oxidative injury by scavenging free radicals and reducing cell oxidative damage. In the study, we measured the activity of antioxidant enzymes such as SOD and POD in transgenic lines and WT under salt stress. Under normal conditions, the activities of SOD and POD in transgenic lines and WT were similar at relatively low levels (Figure 6G,H). However, with the treatment of 150 mM NaCl, the SOD activity in transgenic lines increased significantly by 1.5-fold, 1.33-fold, and 1.49-fold (Figure 6G), respectively, compared to WT plants. And the values of the POD activity were significantly increased by 7.49%, 7.13%, and 8.54%, respectively (Figure 6H).

### 3.8. Histochemical Staining with NBT and DAB in Transgenic Poplars

The presence of ROS facilitates the reduction in nitroblue tetrazolium into a blue formazan. On the other hand, superoxide dismutase scavenges superoxide radicals, inhibiting the formazan’s development [24]. Under the influence of hydrogen peroxide, DAB undergoes an electron loss process, which induces a color change and results in the precipitation of a brown, insoluble material [25]. In the study, we conducted histochemical staining with NBT and DAB in the leaves of transgenic poplars (Figure 7A). The results from staining demonstrated no notable discrepancy in ROS accumulation between transgenic poplars and WT without stress. However, after salt stress, compared to the WT, the transgenic poplar had a notably diminished accumulation of ROS. The results indicated that the transgenic lines had significantly lower O^2−^ accumulation than that in WT, suggesting that the overexpression of *PagbHLH35* has a positive regulatory role in salt tolerance.

### 3.9. Relative Expression Levels of POD and SOD-Related Genes in Transgenic Poplars

To further verify whether *PagbHLH35* regulates plant salt tolerance through the ROS pathway, we used RT-qPCR to measure the relative expression levels of POD and SOD-related genes in transgenic poplar lines. As shown in Figure 7B, under the control condition, there was an absence of notable variation in the expression levels of POD and SOD genes when comparing the transgenic poplar to the WT. Facing salt stress, the relative expression levels of POD-related genes in the transgenic lines were significantly higher than those in the WT, with the lowest increase of 33.33%. However, the relative expression levels of SOD-related genes in the overexpressing lines did not differ significantly from those in the WT, showing only a 1.07-fold increase (Figure 7B). These results suggest that *PagbHLH35* may regulate the expression of POD-related genes, thereby enhancing plant salt tolerance.

## 4. Discussion

### 4.1. Overexpression of PagbHLH35 Enhances Salt Tolerance

When plants are subjected to stress, chlorophyll content serves as direct evidence reflecting salt tolerance. Under salt stress, the chlorophyll concentration in the transgenic poplars that overexpress *PagbHLH35* was considerably elevated compared to the WT. Similarly, *CabHLH35* is the homologous gene of PagbHLH35 in pepper, whose overexpression leads to an increase in chlorophyll content [26]. Proline plays an important role in balancing osmotic pressure, stabilizing biomolecular structures, and regulating cellular redox potential [27]. Under salt stress, the chlorophyll concentration in the transgenic poplars that overexpress PagbHLH35 was considerably elevated compared to the WT. The bHLH TF, *AtMYC2*, participates in the response to salt stress by affecting the expression of the key enzyme *AtP5CS1* in the proline synthesis pathway [28]. Based on the above, the *PagbHLH35* gene positively regulates salt stress tolerance by upregulating chlorophyll content and proline content in the transgenic poplars.

### 4.2. Overexpression of PagbHLH35 Decreased ROS Levels

ROS are chemically active molecules generated within cells [29]. In plants, ROS act as signaling molecules at appropriate levels, participating in the regulation of plant growth, development, and responses to adversity [30]. When subjected to oxidative stress, plants increase resistance by eliminating excess ROS from the body [30]. *NtbHLH123* acts as a molecular switch in the response to salt stress in tobacco, regulating ROS production dependent on oxidase homologs; under salt stress, *bHLH12* activates the genes involved in regulating salt tolerance and increases the levels of the two major antioxidant enzymes superoxide dismutase (SOD) and peroxidase (POD) involved in ROS clearance [31]. In this study, the overexpression of *PagbHLH35* led to lower ROS accumulation, accompanied by the downregulation of MDA content. In conclusion, *PagbHLH35* effectively reduces ROS accumulation under salt stress.

SOD and POD constitute the antioxidant defense system within plants [32] and their synergistic action helps maintain the balance of H_2_O_2_ within cells, preventing oxidative damage caused by excessive accumulation [33]. Studies have shown that the overexpression of *AhbHLH121* promotes the expression of active antioxidant enzymes such as POD and SOD under salt stress, enhancing the ability to scavenge ROS [34]. In our study, the overexpression of *PagbHLH35* resulted in significantly lower levels of hydrogen peroxide under salt stress, suggesting that this gene participates in salt stress regulation, leading to reduced oxidative damage to cell membranes.

### 4.3. PagbHLH35 May Regulate Target Genes through Specially Binding to G-Box/E-Box

In general, bHLH TFs regulate the transcription of target genes by binding to G-box/E-box or GCG-box elements in their promoters, participating in the ABA and Mitogen-Activated Protein Kinase (MAPK) signaling pathways. For example, *TabHLH1* can enhance salt tolerance by mediating the ABA pathway in wheat [35]. *bHLH122* regulates downstream target genes by recognizing G-box (CACGTG) and/or E-box (CANNTG) elements. The overexpression of *bHLH122* in transgenic plants confers increased resistance to drought, salt, and osmotic stress [14]. *AtbHLH112* regulates gene expression by binding to E-box elements and GCG elements, leading to ROS scavenging, ultimately regulating plant stress resistance [36]. In this study, *PagbHLH35* can especially recognize G-box (CACGTG) and/or E-box (CANNTG) elements, as revealed by yeast one-hybrid assays. In addition, it exhibits a significantly upregulated expression of the genes related to the antioxidant system in transgenic poplars, particularly POD-related genes. In conclusion, *PagbHLH35* may regulate the expression of antioxidant enzymes genes by specially binding to G-box and/or E-box in the promoters of downstream genes.

## 5. Conclusions

Under NaCl stress, transgenic poplar overexpressing *PagbHLH35* exhibits significant physiological advantages, including higher activities of peroxidase (POD) and superoxide dismutase (SOD), and further increases chlorophyll and proline content. They also show a lower rate of dehydration, malondialdehyde (MDA) content, and hydrogen peroxide (H_2_O_2_) levels, compared to the wild-type (WT). Histological staining indicates that genetically modified poplars accumulate less reactive oxygen species (ROS) under salt stress. The relative expression levels of several antioxidant genes were significantly higher in genetically modified poplars than those in WT. This suggests that *PagbHLH35* enhances the salt tolerance of genetically modified poplars by improving their ability to scavenge ROS.

## Figures and Tables

**Figure 1 plants-13-01835-f001:**
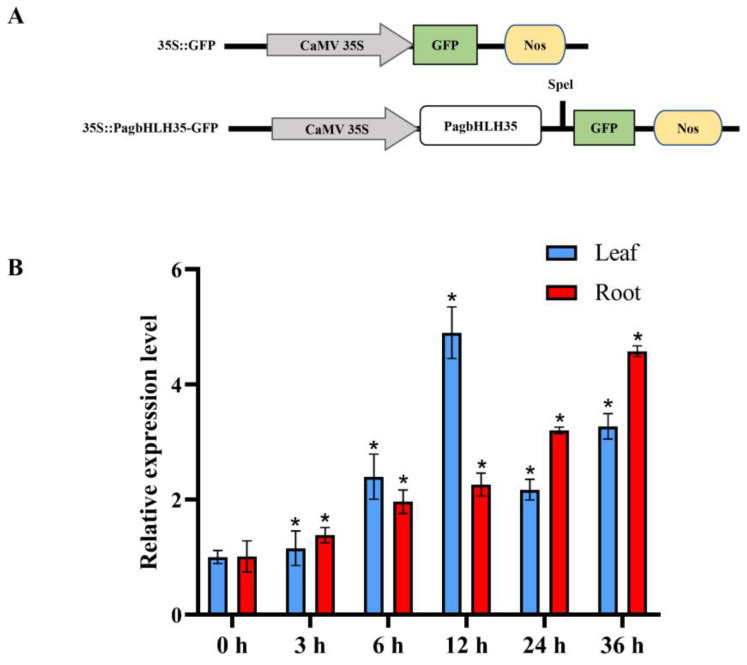
PBI121-PagbHLH35-GFP vector structure and spatiotemporal expression pattern of PagbHLH35 gene. (**A**) PBI121-PagbHLH35-GFP vector structure. (**B**) Spatiotemporal expression pattern of PagbHLH35 gene. Testing method: *t*-test. Asterisks denote significant differences: * *p* ≤ 0.05. Error bars represent standard error for three biological replicates.

**Figure 2 plants-13-01835-f002:**
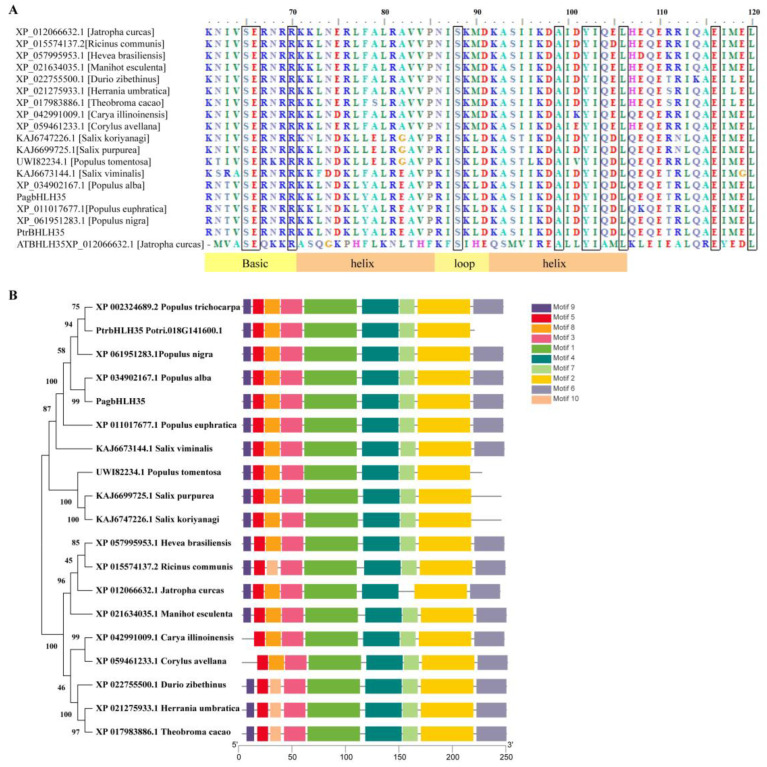
Sequence analysis of PagbHLH35 protein. (**A**) Protein alignment of PagbHLH35 and its homologous proteins from other plant species. (**B**) Motif analysis of BHLH proteins. The clustering was displayed according to phylogenetic tree. MEGA11 was used to build a tree showing how closely related similar proteins are. The neighbor-joining method was used and the tree’s accuracy was checked with 1000 tests.

**Figure 3 plants-13-01835-f003:**
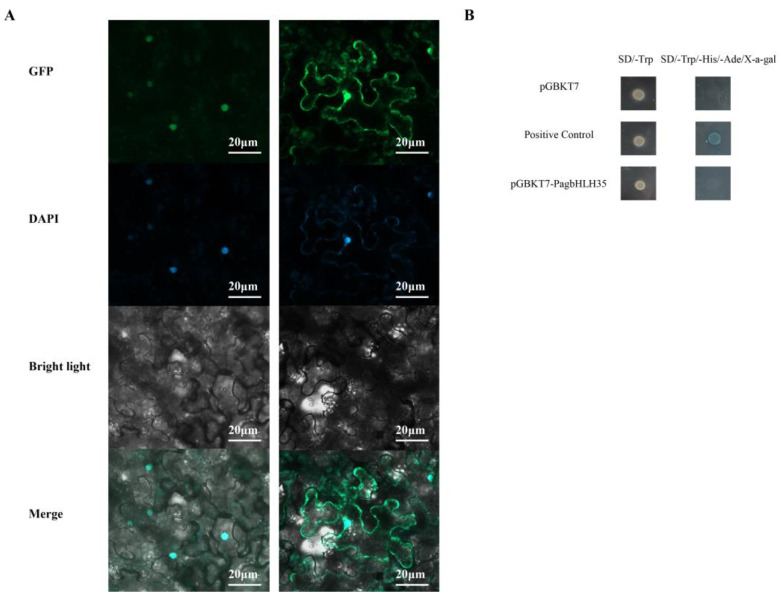
Subcellular localization and transcriptional activity analysis of PagbHLH35 protein. (**A**) Fluorescence observation of PagbHLH35 protein. (GFP) are dark-field images, (DAPI) are dark-field images, (Bright) are bright-field images and (Merged) are superimposed images of dark-field and bright-field. Scale bar = 20 µm. (**B**) Transcriptional activity of PagbHLH35.

**Figure 4 plants-13-01835-f004:**
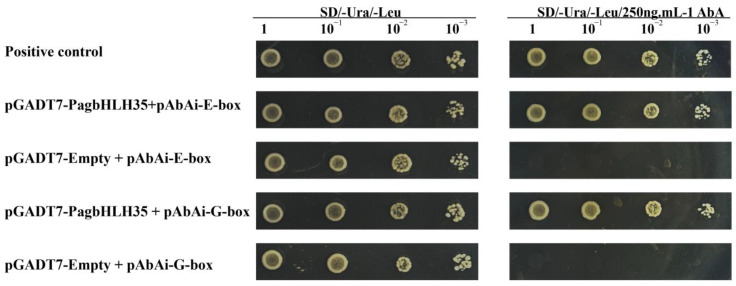
PagbHLH35 specifically binding to G-box and E-box.

**Figure 5 plants-13-01835-f005:**
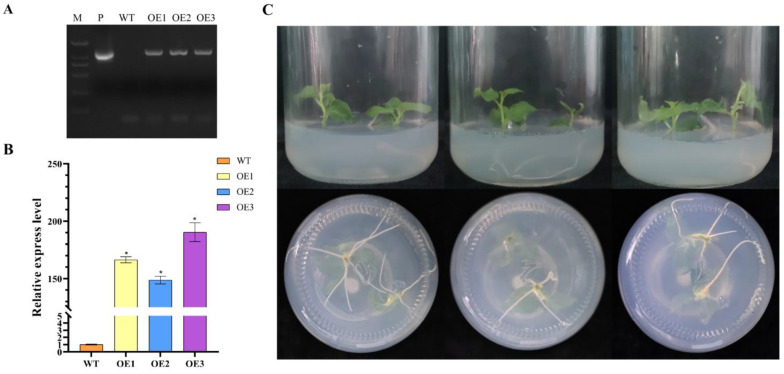
Identification of transgenic poplar lines. WT, wild-type; OE1–3, the transgenic poplar lines overexpressing *PagbHLH35*. (**A**) PCR validation of transgenic poplar lines at DNA level; M, DNA2000 marker; P, positive control. (**B**) RT-qPCR validation of transgenic poplar lines. (**C**) Morphological phenotype of transgenic poplar lines. (Testing method: *t*-test. Asterisks denote significant differences: * *p* ≤ 0.05. Error bars represent standard error for three replicates).

**Figure 6 plants-13-01835-f006:**
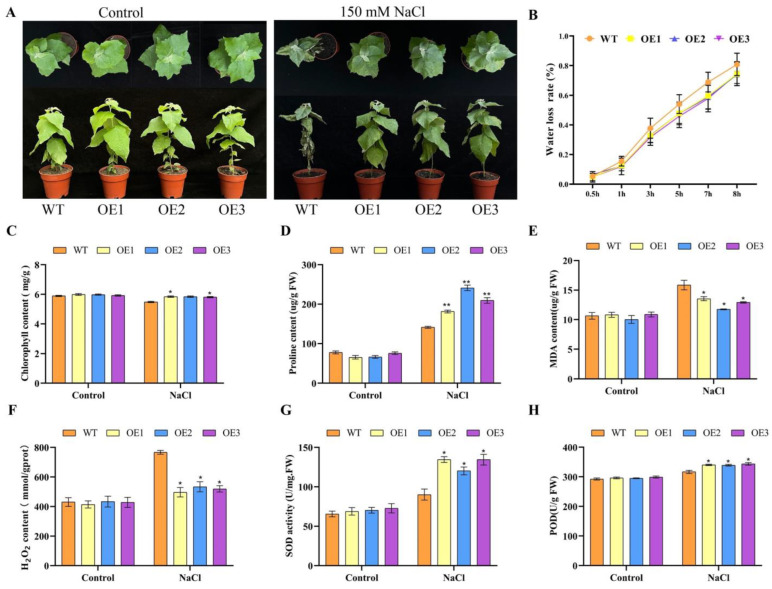
Morphological and physiological characteristics of transgenic poplars under salt stress. (**A**) Morphological phenotypes of transgenic poplars. (**B**) Water loss rate. (**C**) Chlorophyll content. (**D**) Proline content. (**E**) MDA content. (**F**) H_2_O_2_ content. (**G**) SOD activity. (**H**) POD activity. (Testing method: *t*-test. Asterisks denote significant differences: * *p* ≤ 0.05, ** *p* ≤ 0.01 Error bars represent standard error for three replicates).

**Figure 7 plants-13-01835-f007:**
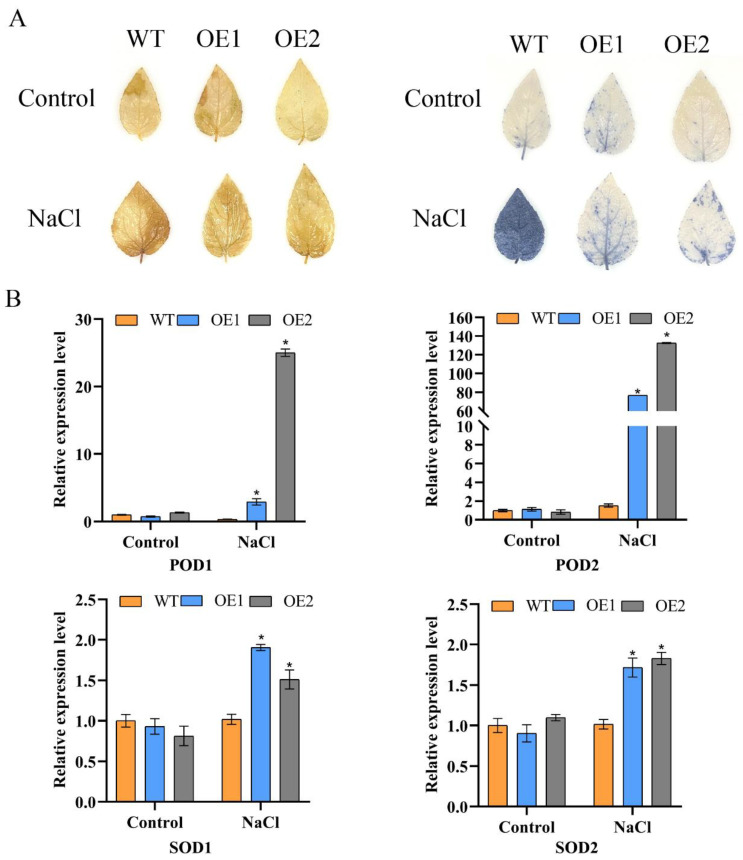
ROS scavenging assay and the expression analysis of antioxidant genes. (**A**) NBT and DAB staining were conducted to detect the content of hydrogen peroxide. (**B**) Relative expression analysis of SODs and PODs. Asterisks indicate significant differences between OE lines and WT (testing method: *t*-test. Asterisks denote significant differences: * *p* ≤ 0.05. Error bars represent standard error for three replicates). WT, wild-type plants; OE1-OE2, the transgenic poplar lines overexpressing *PagbHLH35*.

## Data Availability

Data is contained within the article.

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
