# Peer review of "PagbHLH35 Enhances Salt Tolerance through Improving ROS Scavenging in Transgenic Poplar"

_plants, 2024, doi:10.3390/plants13131835_

Round 1
Reviewer 1 Report
Comments and Suggestions for Authors
Dear Authors,
As per the extensive study of submitted manuscript, you need to improve below queries from my side.
1. Please check extensive English language in text. For your instance L-67, is should start with sentence not with ''And''.
2. Why have you selected 150mM NaCl concentration? why not below or above of mentioned concentration? Why did not analyzed with higher concentration of salinity?
3. Agrobacterium should be italics, please check it.
4. In figure 6, Line 258, H2O2 should be H2O2.
5. In Line 291, why you highlighted the Figure 7B?
6. Did you identify any base pairs from G-box/E-box after interaction?
Please thoroughly check the manuscript, it filled with various spelling mistakes. It needs extensive attention while resubmitting.
Comments on the Quality of English LanguageThis manuscript needs extensive English correction.
Author Response
Dear professor
First and foremost, I would like to express my sincere gratitude for the time and effort you have dedicated to reviewing my manuscript.
Comments 1: Please check extensive English language in text. For your instance L-67, is should start with sentence not with ''And''.
Response 1: Thank you for pointing this out. I agree with this comment. I have removed the word "and" from lines 82 and 152. I correct it to ‘There was lower ROS accumulation detected in the transgenic poplars under salt stress’.I have already corrected that‘ These constructs were co-transformed into Y1H yeast, and positive yeast colonies were identified on SD/-Leu medium and further confirmed on SD/-Leu/AbA medium with serial dilutions at 10x, 100x, and 1000x’.
Comments 2: Why have you selected 150mM NaCl concentration? why not below or above of mentioned concentration? Why did not analyzed with higher concentration of salinity?
Response 2:Thank you for pointing this out. I have collected a lot of literature in the early stage, all of them are salt stress treatment 150mM, I think this concentration is enough to make a stress treatment.
References:1. Zhang, X.; Cheng, Z.; Yao, W.; Gao, Y.; Fan, G.; Guo, Q.; Zhou, B.; Jiang, T. Overexpression of PagERF072 from Poplar Improves Salt Tolerance. Int. J. Mol. Sci. 2022, 23, 10707, doi: 10.3390/ijms231810707.
2.Zhao, X.; Wang, Q.; Yan, C.; Sun, Q.; Wang, J.; Li, C.; Yuan, C.; Mou, Y.; Shan, S. The bHLH transcription factor AhbHLH121 improves salt tolerance in peanut. Int. J. Biol. Macromol. 2024, 256, 128492, doi: 10.1016/j.ijbiomac.2023.128492.
Comments 3: Agrobacterium should be italics, please check it.
Response 3: Thank you for pointing this out. I agree with this suggestion. I have changed "Agrobacterium" in lines 127 and 157 to italics.
Comments 4: In figure 6, Line 258, H2O2 should be H2O2.
Response 4: Thank you for pointing this out. I agree with this suggestion.
Thank you for your correction, I agree with your suggestion, I have already corrected "H2O2" in Figure 6 to "H2O2".
Comments 5: In Line 291, why you highlighted the Figure 7B?
Response 5: Thank you for pointing this out. I have removed the highlighting from Figure 7B in line 367.
Comments 6: Did you identify any base pairs from G-box/E-box after interaction?
Response 6: Thank you for pointing this out. I agree with this comment. We performed a mutation on the component at that time, and the result showed that the interaction disappeared after the mutation.
Once again, thank you for your meticulous review and constructive feedback. I am confident that with your guidance, our article will be more refined and make a more significant contribution to the academic community.
Should you have any further suggestions or require additional information from me, please do not hesitate to contact me.
Reviewer 2 Report
Comments and Suggestions for Authors
The paper is interesting, the research is well conducted and the results are robust, since authors have used three different tree transgenic lines, and performed several different analysis on them, thus strongly supporting the conclussions.
I have only found several minor points:
Fig1: gene name in italics. Please mentioned which statistical test hhave you applied, Please consider this comment for all the figures where the statistical test and the levels of confidence are not explicitly described.
Figure 2: lettering too small.
Figure 3: Enlarge lettering. Center the title. The microscopy requires a nuclear marker such as DAPI to confirm the nuclear localization.
Author Response
Dear professor
First and foremost, I would like to express my sincere gratitude for the time and effort you have dedicated to reviewing my manuscript.
Comments 1: gene name in italics. Please mentioned which statistical test hhave you applied, Please consider this comment for all the figures where the statistical test and the levels of confidence are not explicitly described.
Response 1: Thank you for pointing this out. I agree with this comment. Thank the expert for the valuable suggestions. I have changed the gene names to italics . I have Figure 5and 7 annotated PagbHLH35 changed to italics. The statistical method we used is the t-test. I have annotated figures 1, 5, 6, and 7 with the test method as "t-test."
Comments 2: lettering too small
Response 2: Thank you for pointing this out. The text has been adjusted to size 10 Palatino Linotype as required by the journal.
Comments 3: Enlarge lettering. Center the title. The microscopy requires a nuclear marker such as DAPI to confirm the nuclear localization.
Response 3: Thank you for pointing this out. I agree with this comment. I have centered the title. Regarding the issue that the nuclear localization cannot be determined without DAPI staining, I have conducted additional experiments and updated Figure 3B.
Once again, thank you for your meticulous review and constructive feedback. I am confident that with your guidance, our article will be more refined and make a more significant contribution to the academic community.